# The Power of Asymmetry in Binary Hashing

**Behnam Neyshabur**     **Payman Yadollahpour**     **Yury Makarychev**
Toyota Technological Institute at Chicago
[btavakoli,pyadolla,yury]@ttic.edu

**Ruslan Salakhutdinov**                          **Nathan Srebro**
Departments of Statistics and Computer Science        Toyota Technological Institute at Chicago
University of Toronto                                 and Technion, Haifa, Israel
rsalakhu@cs.toronto.edu                              nati@ttic.edu

## Abstract

When approximating binary similarity using the hamming distance between short binary hashes, we show that even if the similarity is symmetric, we can have shorter and more accurate hashes by using two distinct code maps. I.e. by approximating the similarity between $x$ and $x'$ as the hamming distance between $f(x)$ and $g(x')$, for two distinct binary codes $f, g$, rather than as the hamming distance between $f(x)$ and $f(x')$.

## 1 Introduction

Encoding high-dimensional objects using short binary hashes can be useful for fast approximate similarity computations and nearest neighbor searches. Calculating the hamming distance between two short binary strings is an extremely cheap computational operation, and the communication cost of sending such hash strings for lookup on a server (e.g. sending hashes of all features or patches in an image taken on a mobile device) is low. Furthermore, it is also possible to quickly look up nearby hash strings in populated hash tables. Indeed, it only takes a fraction of a second to retrieve a shortlist of similar items from a corpus containing billions of data points, which is important in image, video, audio, and document retrieval tasks [11, 9, 10, 13]. Moreover, compact binary codes are remarkably storage efficient, and allow one to store massive datasets in memory. It is therefore desirable to find short binary hashes that correspond well to some target notion of similarity. Pioneering work on Locality Sensitive Hashing used random linear thresholds for obtaining bits of the hash [1]. Later work suggested learning hash functions attuned to the distribution of the data [15, 11, 5, 7, 3]. More recent work focuses on learning hash functions so as to optimize agreement with the target similarity measure on specific datasets [14, 8, 9, 6] . It is important to obtain accurate and *short* hashes—the computational and communication costs scale linearly with the length of the hash, and more importantly, the memory cost of the hash table can scale exponentially with the length.

In all the above-mentioned approaches, similarity $S(x, x')$ between two objects is approximated by the hamming distance between the outputs of the same hash function, i.e. between $f(x)$ and $f(x')$, for some $f \in \{\pm 1\}^k$. The emphasis here is that the same hash function is applied to both $x$ and $x'$ (in methods like LSH multiple hashes might be used to boost accuracy, but the comparison is still between outputs of the same function).

The only exception we are aware of is where a single mapping of objects to fractional vectors $\tilde{f}(x) \in [-1,1]^k$ is used, its thresholding $f(x) = \operatorname{sign} \tilde{f}(x) \in \{\pm 1\}^k$ is used in the database, and similarity between $x$ and $x'$ is approximated using $\left\langle f(x), \tilde{f}(x') \right\rangle$. This has become known as "asymmetric hashing" [2, 4], but even with such a-symmetry, both mappings are based on the

same fractional mapping $\tilde{f}(\cdot)$. That is, the asymmetry is in that one side of the comparison gets thresholded while the other is fractional, but not in the actual mapping.

In this paper, we propose using two *distinct* mappings $f(x), g(x) \in \{\pm 1\}^k$ and approximating the similarity $S(x, x')$ by the hamming distance between $f(x)$ and $g(x')$. We refer to such hashing schemes as "asymmetric". Our main result is that even if the target similarity function is symmetric and "well behaved" (e.g., even if it is based on Euclidean distances between objects), using asymmetric binary hashes can be much more powerful, and allow better approximation of the target similarity with shorter code lengths. In particular, we show extreme examples of collections of points in Euclidean space, where the neighborhood similarity $S(x, x')$ can be realized using an asymmetric binary hash (based on a pair of distinct functions) of length $O(r)$ bits, but where a symmetric hash (based on a single function) would require at least $\Omega(2^r)$ bits. Although actual data is not as extreme, our experimental results on real data sets demonstrate significant benefits from using asymmetric binary hashes.

Asymmetric hashes can be used in almost all places where symmetric hashes are typically used, usually without any additional storage or computational cost. Consider the typical application of storing hash vectors for all objects in a database, and then calculating similarities to queries by computing the hash of the query and its hamming distance to the stored database hashes. Using an asymmetric hash means using different hash functions for the database and for the query. This neither increases the size of the database representation, nor the computational or communication cost of populating the database or performing a query, as the exact same operations are required. In fact, when hashing the entire database, asymmetric hashes provide even more opportunity for improvement. We argue that using two different hash functions to encode database objects and queries allows for much more flexibility in choosing the database hash. Unlike the query hash, which has to be stored compactly and efficiently evaluated on queries as they appear, if the database is fixed, an arbitrary mapping of database objects to bit strings may be used. We demonstrate that this can indeed increase similarity accuracy while reducing the bit length required.

## 2  Minimum Code Lengths and the Power of Asymmetry

Let $S : \mathcal{X} \times \mathcal{X} \rightarrow \{\pm 1\}$ be a binary similarity function over a set of objects $\mathcal{X}$, where we can interpret $S(x, x')$ to mean that $x$ and $x'$ are "similar" or "dissimilar", or to indicate whether they are "neighbors". A symmetric binary coding of $\mathcal{X}$ is a mapping $f : \mathcal{X} \rightarrow \{\pm 1\}^k$, where $k$ is the bit-length of the code. We are interested in constructing codes such that the hamming distance between $f(x)$ and $f(x')$ corresponds to the similarity $S(x, x')$. That is, for some threshold $\theta \in \mathbb{R}$, $S(x, x') \approx \text{sign}(\langle f(x), f(x') \rangle - \theta)$. Although discussing the hamming distance, it is more convenient for us to work with the inner product $\langle u, v \rangle$, which is equivalent to the hamming distance $d_h(u, v)$ since $\langle u, v \rangle = (k - 2d_h(u, v))$ for $u, v \in \{\pm 1\}^k$.

In this section, we will consider the problem of capturing a given similarity using an arbitrary binary code. That is, we are given the entire similarity mapping $S$, e.g. as a matrix $S \in \{\pm 1\}^{n \times n}$ over a finite domain $\mathcal{X} = \{x_1, \dots, x_n\}$ of $n$ objects, with $S_{ij} = S(x_i, x_j)$. We ask for an encoding $u_i = f(x_i) \in \{\pm 1\}^k$ of each object $x_i \in \mathcal{X}$, and a threshold $\theta$, such that $S_{ij} = \text{sign}(\langle u_i, u_j \rangle - \theta)$, or at least such that equality holds for as many pairs $(i, j)$ as possible. It is important to emphasize that the goal here is purely to approximate the given matrix $S$ using a short binary code—there is no out-of-sample generalization (yet).

We now ask: Can allowing an asymmetric coding enable approximating a symmetric similarity matrix $S$ with a shorter code length?

Denoting $U \in \{\pm 1\}^{n \times k}$ for the matrix whose columns contain the codewords $u_i$, the minimal binary code length that allows exactly representing $S$ is then given by the following matrix factorization problem:

$$k_s(S) = \min_{k, U, \theta} k \quad \text{s.t} \quad U \in \{\pm 1\}^{k \times n} \qquad \theta \in \mathbb{R} \tag{1}$$
$$Y \triangleq U^\top U - \theta \mathbb{1}_n$$
$$\forall_{ij} \ S_{ij} Y_{ij} > 0$$

where $\mathbb{1}_n$ is an $n \times n$ matrix of ones.

We begin demonstrating the power of asymmetry by considering an *asymmetric* variant of the above problem. That is, even if $S$ is symmetric, we allow associating with each object $x_i$ two distinct binary codewords, $u_i \in \{\pm 1\}^k$ and $v_i \in \{\pm 1\}^k$ (we can think of this as having two arbitrary mappings $u_i = f(x_i)$ and $v_i = g(x_i)$), such that $S_{ij} = \text{sign}(\langle u_i, v_j \rangle - \theta)$. The minimal asymmetric binary code length is then given by:

$$k_a(S) = \min_{k,U,V,\theta} k \quad \text{s.t} \quad U, V \in \{\pm 1\}^{k \times n} \qquad \theta \in \mathbb{R} \qquad (2)$$
$$Y \triangleq U^\top V - \theta \mathbb{1}_n$$
$$\forall_{ij} \ S_{ij} Y_{ij} > 0$$

Writing the binary coding problems as matrix factorization problems is useful for understanding the power we can get by asymmetry: even if $S$ is symmetric, and even if we seek a symmetric $Y$, insisting on writing $Y$ as a square of a binary matrix might be a tough constraint. This is captured in the following Theorem, which establishes that there could be an exponential gap between the minimal asymmetry binary code length and the minimal symmetric code length, even if the matrix $S$ is symmetric and very well behaved:

**Theorem 1.** *For any $r$, there exists a set of $n = 2^r$ points in Euclidean space, with similarity matrix $S_{ij} = \begin{cases} 1 & \text{if } \|x_i - x_j\| \leq 1 \\ -1 & \text{if } \|x_i - x_j\| > 1 \end{cases}$, such that $k_a(S) \leq 2r$ but $k_s(S) \geq 2^r/2$*

*Proof.* Let $I_1 = \{1, \ldots, n/2\}$ and $I_2 = \{n/2 + 1, \ldots, n\}$. Consider the matrix $G$ defined by $G_{ii} = 1/2$, $G_{ij} = -1/(2n)$ if $i, j \in I_1$ or $i, j \in I_2$, and $G_{ij} = 1/(2n)$ otherwise. Matrix $G$ is diagonally dominant. By the Gershgorin circle theorem, $G$ is positive definite. Therefore, there exist vectors $x_1, \ldots, x_n$ such that $\langle x_i, x_j \rangle = G_{ij}$ (for every $i$ and $j$). Define

$$S_{ij} = \begin{cases} 1 & \text{if } \|x_i - x_j\| \leq 1 \\ -1 & \text{if } \|x_i - x_j\| > 1 \end{cases}.$$

Note that if $i = j$ then $S_{ij} = 1$; if $i \neq j$ and $(i, j) \in I_1 \times I_1 \cup I_2 \times I_2$ then $\|x_i - x_j\|^2 = G_{ii} + G_{jj} - 2G_{ij} = 1 + 1/n > 1$ and therefore $S_{ij} = -1$. Finally, if $i \neq j$ and $(i, j) \in I_1 \times I_2 \cup I_2 \times I_1$ then $\|x_i - x_j\|^2 = G_{ii} + G_{jj} - 2G_{ij} = 1 + 1/n < 1$ and therefore $S_{ij} = 1$. We show that $k_a(S) \leq 2r$. Let $B$ be an $r \times n$ matrix whose column vectors are the vertices of the cube $\{\pm 1\}^r$ (in any order); let $C$ be an $r \times n$ matrix defined by $C_{ij} = 1$ if $j \in I_1$ and $C_{ij} = -1$ if $j \in I_2$. Let $U = \begin{bmatrix} B \\ C \end{bmatrix}$ and $V = \begin{bmatrix} B \\ -C \end{bmatrix}$. For $Y = U^\top V - \theta \mathbb{1}_n$ where threshold $\theta = -1$ , we have that $Y_{ij} \geq 1$ if $S_{ij} = 1$ and $Y_{ij} \leq -1$ if $S_{ij} = -1$. Therefore, $k_a(S) \leq 2r$.

Now we show that $k_s = k_s(S) \geq n/2$. Consider $Y$, $U$ and $\theta$ as in (1). Let $Y' = (U^\top U)$. Note that $Y'_{ij} \in [-k_s, k_s]$ and thus $\theta \in [-k_s + 1, k_s - 1]$. Let $q = [1, \ldots, 1, -1, \ldots, -1]^\top$ ($n/2$ ones followed by $n/2$ minus ones). We have,

$$0 \leq q^\top Y' q = \sum_{i=1}^{n} Y'_{ii} + \sum_{i,j:S_{ij}=-1} Y'_{ij} - \sum_{i,j:S_{ij}=1, i \neq j} Y'_{ij}$$
$$\leq \sum_{i=1}^{n} k_s + \sum_{i,j:S_{ij}=-1} (\theta - 1) - \sum_{i,j:S_{ij}=1, i \neq j} (\theta + 1)$$
$$= nk_s + (0.5n^2 - n)(\theta - 1) - 0.5n^2(\theta + 1)$$
$$= nk_s - n^2 - n(\theta - 1)$$
$$\leq 2nk_s - n^2.$$

We conclude that $k_s \geq n/2$. $\qquad \square$

The construction of Theorem 1 shows that there *exists* data sets for which an asymmetric binary hash might be much shorter then a symmetric hash. This is an important observation as it demonstrates that asymmetric hashes could be much more powerful, and should prompt us to consider them instead of symmetric hashes. The precise construction of Theorem 1 is of course rather extreme (in fact, the most extreme construction possible) and we would not expect actual data sets to have this exact structure, but we will show later significant gaps also on real data sets.

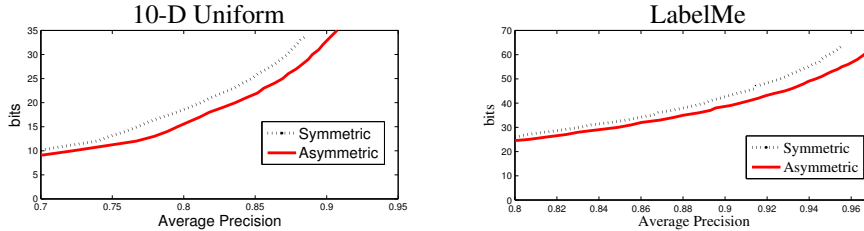

Figure 1: Number of bits required for approximating two similarity matrices (as a function of average precision). Left: uniform data in the 10-dimensional hypercube, similarity represents a thresholded Euclidean distance, set such that 30% of the similarities are positive. Right: Semantic similarity of a subset of LabelMe images, thresholded such that 5% of the similarities are positive.

# 3 Approximate Binary Codes

As we turn to real data sets, we also need to depart from seeking a binary coding that *exactly* captures the similarity matrix. Rather, we are usually satisfied with merely approximating $S$, and for any fixed code length $k$ seek the (symmetric or asymmetric) $k$-bit code that "best captures" the similarity matrix $S$. This is captured by the following optimization problem:

$$\min_{U,V,\theta} \quad L(Y;S) \triangleq \beta \sum_{i,j:S_{ij}=1} \ell(Y_{ij}) + (1-\beta) \sum_{i,j:S_{ij}=-1} \ell(-Y_{ij}) \quad \text{s.t. } U, V \in \{\pm 1\}^{k \times n} \quad \theta \in \mathbb{R} \quad (3)$$
$$Y \triangleq U^\top V - \theta \mathbb{1}_n$$

where $\ell(z) = \mathbf{1}_{z \leq 0}$ is the zero-one-error and $\beta$ is a parameter that allows us to weight positive and negative errors differently. Such weighting can compensate for $S_{ij}$ being imbalanced (typically many more pairs of points are non-similar rather then similar), and allows us to obtain different balances between precision and recall.

The optimization problem (3) is a discrete, discontinuous and highly non-convex problem. In our experiments, we replace the zero-one loss $\ell(\cdot)$ with a continuous loss and perform local search by greedily updating single bits so as to improve this objective. Although the resulting objective (let alone the discrete optimization problem) is still not convex even if $\ell(z)$ is convex, we found it beneficial to use a loss function that is not flat on $z < 0$, so as to encourage moving towards the correct sign. In our experiments, we used the square root of the logistic loss, $\ell(z) = \log^{1/2}(1+e^{-z})$.

Before moving on to out-of-sample generalizations, we briefly report on the number of bits needed empirically to find good approximations of actual similarity matrices with symmetric and asymmetric codes. We experimented with several data sets, attempting to fit them with both symmetric and asymmetric codes, and then calculating average precision by varying the threshold $\theta$ (while keeping $U$ and $V$ fixed). Results for two similarity matrices, one based on Euclidean distances between points uniformly distributed in a hypoercube, and the other based on semantic similarity between images, are shown in Figure 1.

# 4 Out of Sample Generalization: Learning a Mapping

So far we focused on learning binary codes over a fixed set of objects by associating an arbitrary code word with each object and completely ignoring the input representation of the objects $x_i$. We discussed only how well binary hashing can *approximate* the similarity, but did not consider *generalizing* to additional new objects. However, in most applications, we would like to be able to have such an out-of-sample generalization. That is, we would like to learn a mapping $f : \mathcal{X} \to \{\pm 1\}^k$ over an infinite domain $\mathcal{X}$ using only a finite training set of objects, and then apply the mapping to obtain binary codes $f(x)$ for future objects to be encountered, such that $S(x, x') \approx \text{sign}(\langle f(x), f(x') \rangle - \theta)$. Thus, the mapping $f : \mathcal{X} \to \{\pm 1\}^k$ is usually limited to some constrained parametric class, both so we could represent and evaluate it efficiently on new objects, and to ensure good generalization. For example, when $\mathcal{X} = \mathbb{R}^d$, we can consider linear threshold mappings $f_W(x) = \text{sign}(Wx)$, where $W \in \mathbb{R}^{k \times d}$ and $\text{sign}(\cdot)$ operates elementwise, as in Minimal Loss Hashing [8]. Or, we could also consider more complex classes, such as multilayer networks [11, 9].

We already saw that asymmetric binary codes can allow for better approximations using shorter codes, so it is natural to seek asymmetric codes here as well. That is, instead of learning a single

parametric map $f(x)$ we can learn a pair of maps $f : \mathcal{X} \to \{\pm 1\}^k$ and $g : \mathcal{X} \to \{\pm 1\}^k$, both constrained to some parametric class, and a threshold $\theta$, such that $S(x, x') \approx \text{sign}(\langle f(x), g(x') \rangle - \theta)$. This has the potential of allowing for better approximating the similarity, and thus better overall accuracy with shorter codes (despite possibly slightly harder generalization due to the increase in the number of parameters).

In fact, in a typical application where a database of objects is hashed for similarity search over future queries, asymmetry allows us to go even further. Consider the following setup: We are given $n$ objects $x_1, \ldots, x_n \in \mathcal{X}$ from some infinite domain $\mathcal{X}$ and the similarities $S(x_i, x_j)$ between these objects. Our goal is to hash these objects using short binary codes which would allow us to quickly compute approximate similarities between these objects (the "database") and future objects $x$ (the "query"). That is, we would like to generate and store compact binary codes for objects in a database. Then, given a new query object, we would like to efficiently compute a compact binary code for a given query and retrieve similar items in the database very fast by finding binary codes in the database that are within small hamming distance from the query binary code. Recall that it is important to ensure that the bit length of the hashes are small, as short codes allow for very fast hamming distance calculations and low communication costs if the codes need to be sent remotely. More importantly, if we would like to store the database in a hash table allowing immediate lookup, the size of the hash table is exponential in the code length.

The symmetric binary hashing approach (e.g. [8]), would be to find a single parametric mapping $f : \mathcal{X} \to \{\pm 1\}^k$ such that $S(x, x_i) \approx \text{sign}(\langle f(x), f(x_i) \rangle - \theta)$ for future queries $x$ and database objects $x_i$, calculate $f(x_i)$ for all database objects $x_i$, and store these hashes (perhaps in a hash table allowing for fast retrieval of codes within a short hamming distance). The asymmetric approach described above would be to find two parametric mappings $f : \mathcal{X} \to \{\pm 1\}^k$ and $g : \mathcal{X} \to \{\pm 1\}^k$ such that $S(x, x_i) \approx \text{sign}(\langle f(x), g(x_i) \rangle - \theta)$, and then calculate and store $g(x_i)$.

But if the database is fixed, we can go further. There is actually no need for $g(\cdot)$ to be in a constrained parametric class, as we do not need to generalize $g(\cdot)$ to future objects, nor do we have to efficiently calculate it on-the-fly nor communicate $g(x)$ to the database. Hence, we can consider allowing the database hash function $g(\cdot)$ to be an arbitrary mapping. That is, we aim to find a simple parametric mapping $f : \mathcal{X} \to \{\pm 1\}^k$ and $n$ *arbitrary codewords* $v_1, \ldots, v_n \in \{\pm 1\}^k$ for each $x_1, \ldots, x_n$ in the database, such that $S(x, x_i) \approx \text{sign}(\langle f(x), v_i \rangle - \theta)$ for future queries $x$ and for the objects $x_i, \ldots, x_n$ in the database. This form of asymmetry can allow us for greater approximation power, and thus better accuracy with shorter codes, at no additional computational or storage cost.

In Section 6 we evaluate empirically both of the above asymmetric strategies and demonstrate their benefits. But before doing so, in the next Section, we discuss a local-search approach for finding the mappings $f, g$, or the mapping $f$ and the codes $v_1, \ldots, v_n$.

## 5 Optimization

We focus on $x \in \mathcal{X} \subset \mathbb{R}^d$ and linear threshold hash maps of the form $f(x) = \text{sign}(Wx)$, where $W \in \mathbb{R}^{k \times d}$. Given training points $x_1, \ldots, x_n$, we consider the two models discussed above:

**LIN:LIN** We learn two linear threshold functions $f(x) = \text{sign}(W_q x)$ and $g(x) = \text{sign}(W_d x)$. I.e. we need to find the parameters $W_q, W_d \in \mathbb{R}^{k \times d}$.

**LIN:V** We learn a single linear threshold function $f(x) = \text{sign}(W_q x)$ and $n$ codewords $v_1, \ldots, v_n \in \mathbb{R}^k$. I.e. we need to find $W_q \in \mathbb{R}^{k \times d}$, as well as $V \in \mathbb{R}^{k \times n}$ (where $v_i$ are the columns of $V$).

In either case we denote $u_i = f(x_i)$, and in LIN:LIN also $v_i = g(x_i)$, and learn by attempting to minimizing the objective in (3), where $\ell(\cdot)$ is again a continuous loss function such as the square root of the logistic. That is, we learn by optimizing the problem (3) with the additional constraint $U = \text{sign}(W_q X)$, and possibly also $V = \text{sign}(W_d X)$ (for LIN:LIN), where $X = [x_1 \ldots x_n] \in \mathbb{R}^{d \times n}$.

We optimize these problems by alternatively updating rows of $W_q$ and either rows of $W_d$ (for LIN:LIN ) or of $V$ (for LIN:V ). To understand these updates, let us first return to (3) (with un-

constrained $U, V$), and consider updating a row $u^{(t)} \in \mathbb{R}^n$ of $U$. Denote

$$Y^{(t)} = U^\top V - \theta \mathbb{1}_n - u^{(t)\top} v^{(t)},$$

the prediction matrix with component $t$ subtracted away. It is easy to verify that we can write:

$$L(U^\top V - \theta \mathbb{1}_n; S) = C - u^{(t)} M v^{(t)\top} \tag{4}$$

where $C = \frac{1}{2}(L(Y^{(t)} + \mathbb{1}_n; S) + L(Y^{(t)} - \mathbb{1}_n; S))$ does not depend on $u^{(t)}$ and $v^{(t)}$, and $M \in \mathbb{R}^{n \times n}$ also does not depend on $u^{(t)}, v^{(t)}$ and is given by:

$$M_{ij} = \frac{\beta_{ij}}{2} \left( \ell(S_{ij}(Y_{ij}^{(t)} - 1)) - \ell(S_{ij}(Y_{ij}^{(t)} + 1)) \right),$$

with $\beta_{ij} = \beta$ or $\beta_{ij} = (1 - \beta)$ depending on $S_{ij}$. This implies that we can optimize over the entire row $u^{(t)}$ concurrently by maximizing $u^{(t)} M v^{(t)\top}$, and so the optimum (conditioned on $\theta$, $V$ and all other rows of $U$) is given by

$$u^{(t)} = \text{sign}(M v^{(t)}). \tag{5}$$

Symmetrically, we can optimize over the row $v^{(t)}$ conditioned on $\theta$, $U$ and the rest of $V$, or in the case of LIN:V, conditioned on $\theta$, $W_q$ and the rest of $V$.

Similarly, optimizing over a row $w^{(t)}$ of $W_q$ amount to optimizing:

$$\arg \max_{w^{(t)} \in \mathbb{R}^d} \text{sign}(w^{(t)} X) M v^{(t)\top} = \arg \max_{w^{(t)} \in \mathbb{R}^d} \sum_i \left\langle M_i, v^{(t)} \right\rangle \text{sign}(\left\langle w^{(t)}, x_i \right\rangle). \tag{6}$$

This is a weighted zero-one-loss binary classification problem, with targets $\text{sign}(\langle M_i, v^{(t)} \rangle)$ and weights $|\langle M_i, v^{(t)} \rangle|$. We approximate it as a weighted logistic regression problem, and at each update iteration attempt to improve the objective using a small number (e.g. 10) epochs of stochastic gradient descent on the logistic loss. For LIN:LIN, we also symmetrically update rows of $W_d$.

When optimizing the model for some bit-length $k$, we initialize to the optimal $k - 1$-length model. We initialize the new bit either randomly, or by thresholding the rank-one projection of $M$ (for unconstrained $U, V$) or the rank-one projection after projecting the columns of $M$ (for LIN:V) or both rows and columns of $M$ (for LIN:LIN) to the column space of $X$. We take the initialization (random, or rank-one based) that yields a lower objective value.

## 6 Empirical Evaluation

In order to empirically evaluate the benefits of asymmetry in hashing, we replicate the experiments of [8], which were in turn based on [5], on six datasets using learned (symmetric) linear threshold codes. These datasets include: LabelMe and Peekaboom are collections of images, represented as 512D GIST features [13], Photo-tourism is a database of image patches, represented as 128 SIFT features [12], MNIST is a collection of 785D greyscale handwritten images, and Nursery contains 8D features. Similar to [8, 5], we also constructed a synthetic 10D Uniform dataset, containing uniformly sampled 4000 points for a 10D hypercube. We used 1000 points for training and 3000 for testing.

For each dataset, we find the Euclidean distance at which each point has, on average, 50 neighbours. This defines our ground-truth similarity in terms of neighbours and non-neighbours. So for each dataset, we are given a set of $n$ points $x_1, \ldots, x_n$, represented as vectors in $\mathcal{X} = \mathbb{R}^d$, and the binary similarities $S(x_i, x_j)$ between the points, with +1 corresponding to $x_i$ and $x_j$ being neighbors and -1 otherwise. Based on these $n$ training points, [8] present a sophisticated optimization approach for learning a thresholded linear hash function of the form $f(x) = \text{sign}(Wx)$, where $W \in \mathbb{R}^{k \times d}$. This hash function is then applied and $f(x_1), \ldots, f(x_n)$ are stored in the database. [8] evaluate the quality of the hash by considering an independent set of *test points* and comparing $S(x, x_i)$ to $\text{sign}(\langle f(x), f(x_i) \rangle - \theta)$ on the test points $x$ and the database objects (i.e. training points) $x_i$.

In our experiments, we followed the same protocol, but with the two asymmetric variations LIN:LIN and LIN:V, using the optimization method discussed in Sec. 5. In order to obtain different balances between precision and recall, we should vary $\beta$ in (3), obtaining different codes for each value of

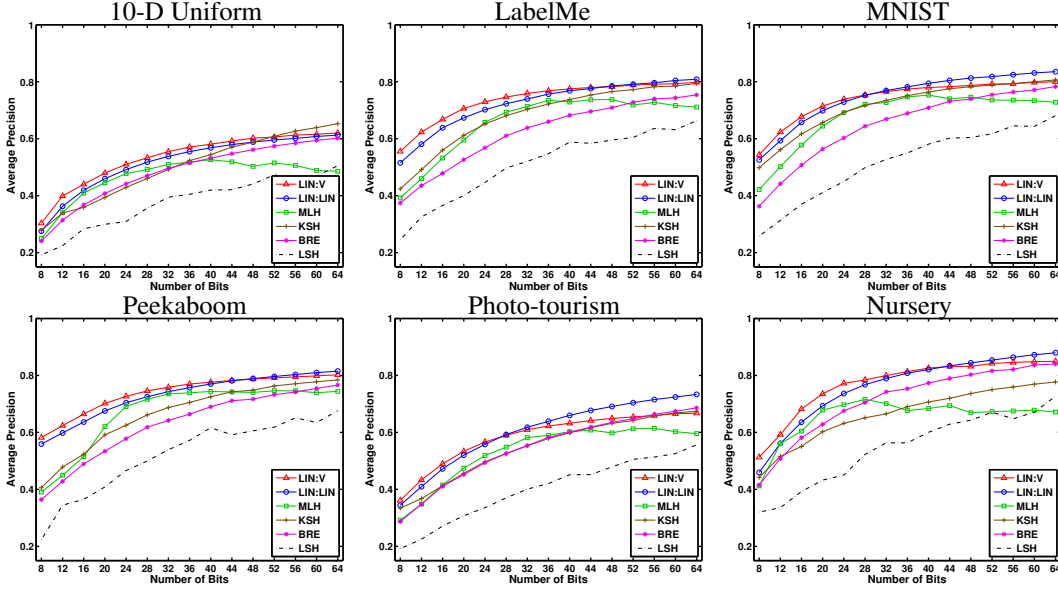

Figure 2: Average Precision (AP) of points retrieved using Hamming distance as a function of code length for six datasets. Five curves represent: LSH, BRE, KSH, MLH, and two variants of our method: Asymmetric LIN-LIN and Asymmetric LIN-V. (Best viewed in color.)

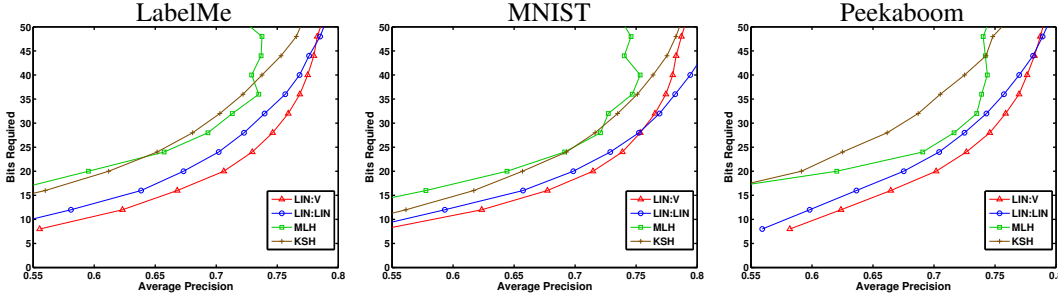

Figure 3: Code length required as a function of Average Precision (AP) for three datasets.

$\beta$. However, as in the experiments of [8], we actually learn a code (i.e. mappings $f(\cdot)$ and $g(\cdot)$, or a mapping $f(\cdot)$ and matrix $V$) using a fixed value of $\beta = 0.7$, and then only vary the threshold $\theta$ to obtain the precision-recall curve.

In all of our experiments, in addition to Minimal Loss Hashing (MLH), we also compare our approach to three other widely used methods: Kernel-Based Supervised Hashing (KSH) of [6], Binary Reconstructive Embedding (BRE) of [5], and Locality-Sensitive Hashing (LSH) of [1]. [1]

In our first set of experiments, we test performance of the asymmetric hash codes as a function of the bit length. Figure 2 displays Average Precision (AP) of data points retrieved using Hamming distance as a function of code length. These results are similar to ones reported by [8], where MLH yields higher precision compared to BRE and LSH. Observe that for all six datasets both variants of our method, asymmetric LIN:LIN and asymmetric LIN:V , consistently outperform all other methods for different binary code length. The gap is particularly large for short codes. For example, for the LabelMe dataset, MLH and KSH with 16 bits achieve AP of 0.52 and 0.54 respectively, whereas LIN:V already achieves AP of 0.54 with only 8 bits. Figure 3 shows similar performance gains appear for a number of other datasets. We also note across all datasets LIN:V improves upon LIN:LIN for short-sized codes. These results clearly show that an asymmetric binary hash can be much more compact than a symmetric hash.

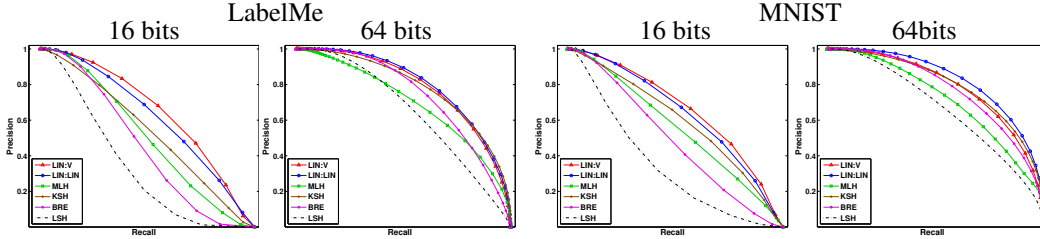

Figure 4: Precision-Recall curves for LabelMe and MNIST datasets using 16 and 64 binary codes. (Best viewed in color.)

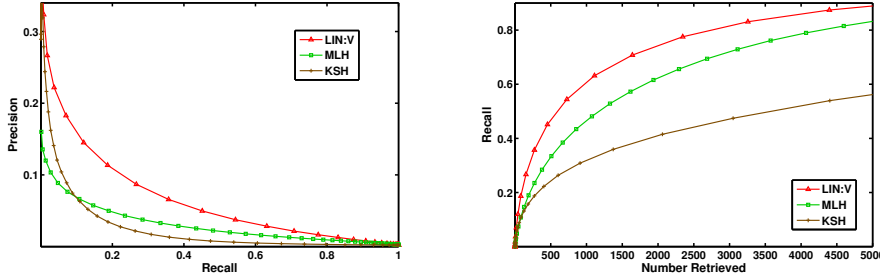

Figure 5: **Left:** Precision-Recall curves for the Semantic 22K LabelMe dataset **Right:** Percentage of 50 ground-truth neighbours as a function of retrieved images. (Best viewed in color.)

Next, we show, in Figure 4, the full Precision-Recall curves for two datasets, LabelMe and MNIST, and for two specific code lengths: 16 and 64 bits. The performance of LIN:LIN and LIN:V is almost uniformly superior to that of MLH, KSH and BRE methods. We observed similar behavior also for the four other datasets across various different code lengths.

Results on previous 6 datasets show that asymmetric binary codes can significantly outperform other state-of-the-art methods on relatively small scale datasets. We now consider a much larger LabelMe dataset [13], called *Semantic 22K LabelMe*. It contains 20,019 training images and 2,000 test images, where each image is represented by a 512D GIST descriptor. The dataset also provides a semantic similarity $S(x, x')$ between two images based on semantic content (object labels overlap in two images). As argued by [8], hash functions learned using semantic labels should be more useful for content-based image retrieval compared to Euclidean distances. Figure 5 shows that LIN:V with 64 bits substantially outperforms MLH and KSH with 64 bits.

## 7 Summary

The main point we would like to make is that when considering binary hashes in order to approximate similarity, even if the similarity measure is entirely symmetric and "well behaved", much power can be gained by considering asymmetric codes. We substantiate this claim by both a theoretical analysis of the possible power of asymmetric codes, and by showing, in a fairly direct experimental replication, that asymmetric codes outperform state-of-the-art results obtained for symmetric codes. The optimization approach we use is very crude. However, even using this crude approach, we could find asymmetric codes that outperformed well-optimized symmetric codes. It should certainly be possible to develop much better, and more well-founded, training and optimization procedures.

Although we demonstrated our results in a specific setting using linear threshold codes, we believe the power of asymmetry is far more widely applicable in binary hashing, and view the experiments here as merely a demonstration of this power. Using asymmetric codes instead of symmetric codes could be much more powerful, and allow for shorter and more accurate codes, and is usually straight-forward and does not require any additional computational, communication or significant additional memory resources when using the code. We would therefore encourage the use of such asymmetric codes (with two distinct hash mappings) wherever binary hashing is used to approximate similarity.

**Acknowledgments**

This research was partially supported by NSF CAREER award CCF-1150062 and NSF grant IIS-1302662.

## Footnotes

[1] We used the BRE, KSH and MLH implementations available from the original authors. For each method, we followed the instructions provided by the authors. More specifically, we set the number of points for each hash function in BRE to 50 and the number of anchors in KSH to 300 (the default values). For MLH, we learn the threshold and shrinkage parameters by cross-validation and other parameters are initialized to the suggested values in the package.

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
