[Reviews · NeurIPS 2013]

Submitted by Assigned_Reviewer_3

This paper proposes a simple modification to the usual "binary hashing for similarity" scheme: use two different maps for the query and the database. The authors then provide compelling theoretical and experimental results showing why using different mapping can yield large benefits in terms of shorter code length and better precision.

Pros:
1. The paper's main idea is simple yet novel, and isn't more demanding computationally than current state-of-the-art methods.
2. The paper gives an easily understandable yet persuasive theoretical argument in favor of using asymmetric hash functions: The argument relies on the relative ease in which a similarity matrix S can be decomposed into the product of two different matrices S~U*V', as opposed to a PSD-like decomposition, S~U*U'.
3. The experimental results are strong. The authors rigorously show significant improvement over state-of-the-art both in terms of precision and in terms of code length for various regimes (short-long codes, medium-high precision) on a variety of datasets.
4. The paper is well written and clear.

Cons:
1. I would have been happy to see a more thorough explanation of the overall optimization scheme, especially for the Lin:V variant.
2. The authors strongly claim that the improvement they show in experiments stems from the asymmetry of the code, and not the optimization method they use (which they term "crude"). However, I believe they use a different optimization scheme from the one used in MLH, BRE or LSH. A few more words on why they believe this so strongly would be of help.


Minor comments:
1. Lines 110-111: The minimum should be over V as well.
2. Line 142: The inequality sign is reversed.
3. Line 143: The vector r could be confused with the r used in n=2^r.
Summary: A strong paper presenting a simple yet novel idea. The authors show theoretically and experimentally the value of using asymmetric hash functions.

Submitted by Assigned_Reviewer_6

This paper proposed to approximate binary similarity using asymmetric binary mappings. The authors empirically show that using asymmetric binary hashes can approximate better the target similarity with shorter code lengths.


This paper is well-written and is easy to follow in most sections. The idea is novel and interesting. Since in order to approximate the binary similarity, there is no requirement to constraint Y_ij in equation (1) to be symmetric, learning asymmetric binary hashes from this view improve the model complexity and thus reasonably expect to use shorter code lengths. However, the authors fail to clarify this point in an intuitive way.

Instead, the authors try to elaborate the asymmetric idea by Theorem 1. However, the proof of Theorem 1 seems to be problematic. For example, according to the equation in line 129-132, the S_ij in line 134 should be 1 rather than -1. Similarly, the S_ij in line 135 should be -1. Furthermore, the lines 142-154 are also difficult to follow.

Finally, in the section of optimization, lines 273-287 are difficult to follow. In general, it would be good if the authors could outline the whole algorithm.
Summary: This paper proposed to approximate binary similarity using asymmetric hash codes. The idea is novel and reasonable. Empirical results also seem to be promising. However, some part of this paper is difficult to follow.

Submitted by Assigned_Reviewer_7

This paper suggested constructing different binary data representation using different hash functions in the asymmetric theme to possibly reduce the encoding length. Encoding data as compactly as possible is no doubt important to theory and practice of approximate nearest neighbor search.

Detailed comments:

1. A major concern of the asymmetric design is the consistency of the results. In Eqns. (2) (3) and Theorem 1, there is no constraint enforcing Y's symmetry, although there was a sentence vaguely mentioning this issue below Eqn. (2). Therefore, by this design it is very possible to obtain inconsistent results such as < u(x), v(y) > ~= < u(y), v(x) >, which means that the learned hash functions CANNOT support a distance metric. And in the actual experiments, the authors used whether a certain point x is in the database to determine which hash function to use, which is ad-hoc and lacks any ground support. Hence, this asymmetric design on this hand makes the theoretical model inconsistent and broken, and on the other hand weakens the experimental evidence, considering that too many parameters were used.

2. Some recent important state-of-the-art hashing methods are missing, e.g.,

Yunchao Gong, S. Lazebnik, A. Gordo, and F. Perronnin. Iterative Quantization: A Procrustean Approach to Learning Binary Codes for Large-scale Image Retrieval. TPAMI 2012.
W. Liu, J. Wang, S. Kumar, and S.-F. Chang. Hashing with Graphs. ICML 2011.
W. Liu, J. Wang, R. Ji, Y.-G. Jiang, and S.-F. Chang. Supervised Hashing with Kernels. CVPR 2012.

The experiments only compared with some baselines known to be problematic in the short-bit setting such as LSH, thus are not too convincing. In addition, kernel-based supervised hashing (the CVPR12 paper mentioned above) has shown superior performance over BRE and MLH. Why not cited and compared? The idea of manipulating code inner products was proposed by this work. The authors should give clear credit to the CVPR12 paper when using code inner products in their method. The authors also need to clarify the relationship and difference with the prior asymmetric hashing work [3] (cite the PAMI version).

3. L084-085: the value for < u, v > seems incorrect.

4. L142-152: aka, second part of proof to Theorem 1; Line 142, it should be ks(S) >= n/2; Line 143, why \theta is in that range? Can the authors explain? Line 146-150: how to move to the upper bound, in particular for the latter two terms? The current derivation is doubtful.

5. Part 4-5: There is an important turn from approximation to generalization, which makes the main thrust of the paper. My major concerns are:
1) Even if we know that compact codes are possible, why is a linear threshold function one appropriate choice for forming hash functions?
2) Given that in advance we do not know the data space much, how to choose k properly in Eqn. (3)? This could be a hard decision in practice. I think that a more pertinent version could fix an approximation tolerance, and minimize k instead. The presentation in Part 4 is loose.

Overall, I think that the asymmetric similarity/distance approximation problem and some results in the paper are interesting. When treating practical problems, the proposed methods (Part 4-5) stay at the heuristic level and do not correspond well to the former claim. The bottom line is that the heuristics are justified by sufficient experiments with positive results. In addition, I think that the authors should put more efforts on polishing their paper, in view of numerous statement problems and grammatical errors.
Summary: In this paper, the authors proposed an asymmetric hashing scheme which uses different hash functions for the database samples and the query sample. By formulating similarity approximation with compact binary codes into an optimization problem that minimizes the bit length with hard constraints imposed on the inner-products among hash codes, the paper then introduced two kinds of hash codes (i.e., two hash functions) to make the constraints asymmetric. The authors then illustrated how to "soften" the constraints to make the asymmetric hashing scheme practical in real applications, and presented how to learn generalizable hash functions. The optimization was done via weighted logistic regression and greedy bitwise updates. Experiments on six datasets showed higher average precision and shorter code length compared with LSH, BRE, and MLH.
Author Feedback

Author rebuttal: We thank all reviewers for their feedback and will incorporate their suggestions. We respond to some specific issues below. The major issues we would like to emphasize are:

Theorem 1 is correct, despite a few confusing typos in the proof-see below.

The approximate similarity matrix Y may be asymmetric (as Rev_7 points out), but we view this as an advantage (because of the added flexibility, which we harness) and claim that in many applications, it is not important that it is not symmetric or cannot support a distance metric--only that it approximates the true similarities well.

We compared to a recent state-of-the-art paper, replicating their methodology. We tried to compare also to Liu et. al. CVPR12, but could not get their code. [UPDATE: we now have the code and do compare to Liu et al]

DETAILED RESPONSE

* Theorem 1 is correct and the statement holds. We apologize for several typos (most of which the reviewers caught!) resulting from last-minute presentation changes:

[all typos were corrected, and the current version is correct]

* Some clarifications regarding optimization:

When optimizing the model for bit-length k, we initialize the algorithm to the optimal k-1 bit-length, add an additional “bit”, and then iteratively try to improve by re-optimizing one of the k “bits” at a time.

Let u and v be rows of matrices U and V corresponding to one “bit” that we are trying to add or improve. To do so, we first form a matrix M as explained in lines 270-282. We now try to maximize uMv^T by iteratively fixing v and optimizing over u and vice versa.

When fixing v, we also fit W_q using Eq. 6. When fixing u, we fit W_d using Eq. 6 (for LIN:LIN variant of the algorithm) or fit the codes v using Eq. 5 (for LIN:V version). We repeatedly continue alternating updates till convergence.

Rev_3 makes a good point, which is that it is hard to tell how much of the empirical improvement is due to the differences in optimization versus the asymmetry. This is a valid point, we are also bothered by it, but it's very hard to address due to the heuristic nature of the optimization, and since different optimization procedures are appropriate for symmetric and asymmetric hashings (our alternating approach is inherently asymmetric). We certainly see empirical benefits with asymmetric codes, and it might well be that some of the benefit is because asymmetric codes are easier to optimize using known methods. We are working on better optimization for both symmetric and asymmetric hashes, which might shed more light on the issue.

* Responses to Rev_7

1) We disagree that the asymmetry of Y breaks our model. The matrix Y can indeed be asymmetric. In fact, the point of this paper is that one can gain a lot by allowing the code (and hence possibly also the matrix Y) to be asymmetric, while in many retrieval and approximate neighbour search we are in any case doing things “approximately” and the goal is to be closest to the truth (the matrix S in our case)--maximize the quality of returned similar objects and minimize the number of full-distance computations on false-positive candidates.

In the retrieval tasks, we always use the first hash function for a query item and a second hash function to generate codes for items in a database. In NN search, if you want to find the top k-NN of item i, take i as query and search for neighbors in the database. Note that the k-NN similarity matrix is anyway asymmetric because if i is a top-k-neighbor of j it doesn’t mean that j is a top-k-neighbor of i.

2) Baseline for comparison: MLH and BRE do have good performance even on short bit setting and this is verified even in the CVPR 2012 mentioned (we include LSH only for reference--beating it is of course not the point here). We also tried to get the source code for Liu et al CVPR 2012. We filled out the online forms and also contacted the author but were not successful in obtaining the code. We did manage getting the code this past week, and will now perform the comparison.

We will cite the PAMI version of [3] but the difference between the definition of asymmetric code in previous works and our definition is clearly stated in lines 46-52.

3) This is a typo. It should read: < u,v > = k-2d(u,v).

4) Theorem 1 is valid, and the proof is correct with the typo corrections above. Regarding lines 143 and 148: Each Y_ij is the product of length k sign vectors, thus Y_ij are integers and all have the same parity as k. Hence we may assume WLOG that the threshold theta is an integer of different parity. Since Y_ij is integer, S_ij Y_ij > 0 from (1) implies S_ij Y_ij > = 1. Now if S_ij = -1 then Y_ij < = -1 and Y’_ij < = theta - 1. Similarly, if S_ij = 1 then Y_ij > = 1 and Y_ij’ > = theta + 1. We use these bounds in line 148. Consider i and j with S_ij =1. We have theta < = Y’_ij -1 < = k_s -1. Now consider i and j with S_ij = -1. We have theta > = Y’_ij + 1 > = - k_s + 1. We proved the bound on k_s in line 143.

5.1) We do not claim that linear threshold functions are the best. Section 4 refers to generic classes of functions and mentions several different classes. We used these in the experiments because it is probably the most widely used parametric class, and because we tried to replicate the MLH experiments as closely as possible. In lines 426-426 we explicitly say we view this only as an initial demonstration.

5.2) Our optimization approach adds bits incrementally. And so, instead of specifying the number of bits, it’s easy to specify the desired accuracy, and then stop adding bits when the desired accuracy is reached.